# PARM1 Drives Smooth Muscle Cell Proliferation in Pulmonary Arterial Hypertension via AKT/FOXO3A Axis

**DOI:** 10.3390/ijms24076385

**Published:** 2023-03-28

**Authors:** Zhen He, Teding Chang, Yu Chen, Hongjie Wang, Lei Dai, Hesong Zeng

**Affiliations:** 1Division of Cardiology, Department of Internal Medicine, Tongji Hospital, Tongji Medical College, Huazhong University of Science and Technology, Wuhan 430030, Chinahongjie.wang@tjh.tjmu.edu.cn (H.W.); 2Hubei Provincial Engineering Research Center of Vascular Interventional Therapy, Wuhan 430030, China; 3Division of Trauma & Surgical Critical Care, Department of Surgery, Tongji Hospital, Tongji Medical College, Huazhong University of Science and Technology, Wuhan 430030, China

**Keywords:** pulmonary arterial hypertension, weighted gene co-expression network analysis (WGCNA), PARM1, AKT, FOXO3A, proliferation

## Abstract

Pulmonary arterial hypertension (PAH) is a group of severe, progressive, and debilitating diseases with limited therapeutic options. This study aimed to explore novel therapeutic targets in PAH through bioinformatics and experiments. Weighted gene co-expression network analysis (WGCNA) was applied to detect gene modules related to PAH, based on the GSE15197, GSE113439, and GSE117261. GSE53408 was applied as validation set. Subsequently, the validated most differentially regulated hub gene was selected for further ex vivo and in vitro assays. *PARM1*, *TSHZ2*, and *CCDC80* were analyzed as potential intervention targets for PAH. Consistently with the bioinformatic results, our ex vivo and in vitro data indicated that PARM1 expression increased significantly in the lung tissue and/or pulmonary artery of the MCT-induced PAH rats and hypoxia-induced PAH mice in comparison with the respective controls. Besides, a similar expression pattern of PARM1 was found in the hypoxia- and PDGF--treated isolated rat primary pulmonary arterial smooth muscle cells (PASMCs). In addition, hypoxia/PDGF--induced PARM1 protein expression could promote the elevation of phosphorylation of AKT, phosphorylation of FOXO3A and PCNA, and finally the proliferation of PASMCs in vitro, whereas PARM1 siRNA treatment inhibited it. Mechanistically, PARM1 promoted PAH via AKT/FOXO3A/PCNA signaling pathway-induced PASMC proliferation.

## 1. Introduction

Pulmonary arterial hypertension (PAH) is a collection of progressive and devastating diseases with limited therapeutic options [1,2,3]. PAH is physiologically defined as a resting mean pulmonary artery pressure of at least 20 mm Hg, pulmonary vascular resistance of more than 3 Wood units, and a pulmonary capillary wedge pressure of less than 15 mm Hg [4]. Although considerable effort has been expended to explore effective intervention targets for PAH, the current therapies focusing on pulmonary vasoconstriction remain unsatisfactory, due to poor long-term outcomes [5,6,7]. Vascular remodeling represents a complex process that is orchestrated by dysfunction of the pulmonary vascular endothelial cells, the pathological proliferation and migration of the pulmonary arterial smooth muscle cells (PASMCs), thrombosis, and inflammation [8,9,10,11]. It has been widely accepted that PASMC proliferation plays a crucial role in pulmonary vascular remodeling. Thus, an effective strategy targeting the basic mechanism of vascular remodeling, including the proliferation and migration of PASMCs, could represent a breakthrough in PAH treatment. 

As the omics field grows, its application in the development of new tools and analysis techniques enables its use in translational medicine through integrating novel knowledge [12,13,14,15]. Weighted gene co-expression network analysis (WGCNA), a promising bioinformatic method for constructing co-expression networks based on gene expression data profiles, provides novel insights for finding key regulators of diseases [16,17,18]. 

Herein, a WGCNA was implemented using gene expression data from several public microarray datasets to identify hub genes related to PAH, of which prostate androgen-regulated mucin-like protein 1 (PARM1) was selected for subsequent functional assays ex vivo and in vitro. Our findings provide novel clues for the further clinical development of PARM1 as a therapeutic target for PAH.

## 2. Results

### 2.1. Data Processing and DEGs Screening

The combined PAH dataset containing three whole-transcriptome sequencing datasets (GSE15197, GSE113439, GSE117261) includes 148 patients (99 PAH, 49 non-PAH) and 17,664 genes for each patient. To verify the expression of these genes between PAH and control patients, we performed a DEG analysis on the combined gene expression dataset, which identified 1,557 upregulated and 1,848 downregulated genes (Figure 1A). A heat map shows the upregulated and downregulated genes in PAH and normal lungs (Figure 1B). The biological significance of these genes was explored through a GO gene set enrichment analysis, and we found that the signaling and signal transduction was reduced in PAH lungs, while the response to the stimulus was increased (Figure 1C). These results showed that, in the PAH lungs, the signal transduction was impaired, and these lung cells were more stressed because they were devoid of oxygen. The main biological process of PAH occurred in the extracellular space and plasma membrane. The cell communication was also decreased in the PAH lungs.

### 2.2. Hierarchical Clustering Analysis, GO, and Pathway Enrichment Analysis

These genes were clustered using WGCNA, which is a network analysis tool used to determine the most correlated genes and cluster them into one module. The soft-threshold power (β), which is a logarithmic scale proportional to the similarity, was critical for network construction (Appendix A). When the β was increased, the scale topology model fit also increased, until the model tended to be stable. The first β that made the scale topology model fit stable was the cut-off that we wanted. In this experiment, the β value was 6. In the conditions of β = 6, WGCNA identified 42 modules, which were randomly assigned color labels by the dynamic (Figure 2A). According to the module–trait relationships between modules and PAH traits (Figure 2B,C), the white module was selected, due to its significant positive correlation with PAH (r = 0.5, *p* < 0.01). Module membership (MM), a “fuzzy” measure of module membership correlating gene expression profile with the module eigengene of a given module, was positively correlated with GS values, which meant that the genes that were positively correlated with the module eigengene would also be positive with GS. The MM and GS correlation combined the genes and PAH, which gave a basis for determining the hub genes (Figure 2D). The gene expression heat map shows that the white module genes were upregulated in PAH patients (Appendix A). The biological significance of the white module was explored through GO enrichment analysis (Figure 2E). The extracellular matrix organization and external encapsulating structure organization of biological pathways (BP) were enriched in the PAH patients. Collagen-containing extracellular matrix and glycosaminoglycan binding were also enriched in PAH patients. The white module represented the extracellular cell activity and was mainly concentrated on the molecular binding pathways.

### 2.3. CCDC80, PARM1, and TSHZ2: Potential Intervention Targets for PAH

The gene network of the white module was constructed using a topological overlap matrix (TOM) with a threshold = 0.02 (Figure 3A). This network was composed of 90 genes, and each gene was connected with the others. The top genes were *CCDC80*, *PAMR1*, and *TSHZ2*. *CCDC80*, which was located in the extracellular matrix, enables glycosaminoglycan binding activity and acts upstream of, or within, an extracellular matrix organization. The cell–substrate adhesion is regulated by *CCDC80* [19]. It has been reported that *CCDC80* may be a novel biomarker and therapeutic target in PAH [20]. *TSHZ2* encoded a protein with five C2H2-type zinc fingers, a homeobox DNA-binding domain, and a coiled-coil domain [19]. This gene acts as a transcriptional repressor and plays a role in the development and progression of cancer [21]. Although *TSHZ2* and *CCDC80* had a powerful discrimination ability (AUC_TSHZ2_ = 0.977; AUC_CCDC80_ = 0.833) in GSE53408 (Figure 3B), we did not select these two genes, due to the sequence uncertainty of *TSHZ2* and the certainty of the correlation between *CCDC80* and PAH. *PARM1*, which was located in the extracellular region [22], enables calcium ion binding activity and serine-type endopeptidase activity [23,24], and it also had an excellent diagnostic power to distinguish the normal and PAH lungs (AUC_PAMR1_ = 0.795). *CCDC80*, *TSHZ2*, and *PARM1* were upregulated in the PAH lungs (Figure 3C). The decision curve analysis graphically shows the clinical usefulness of each gene based on a continuum of potential thresholds (x-axis) and the net benefit of using the genes to risk PAH patients (y-axis). Thus, it was used to facilitate the comparison between the above-mentioned 3 genes. In the analysis, the *PAMR1* and *TSHZ2* provided a larger net benefit across the range of risk thresholds compared with the *CCDC80* (Figure 3D). However, we did not find an increase of *PARM1*, *CCDC80*, and *TSHZ2* in blood samples of PAH patients, which means these genes were non-secreted factors located in the lungs and induced the progression of PAH.

### 2.4. Establishment of PAH Rodents

PAH rodents were established after receiving a single dose of 40 mg/kg (Intraperitoneal) MCT or exposure to chronic hypoxia, respectively (Figure 4A and Figure 5A). The mean right ventricle systolic pressure (RVSP) was significantly higher in the monocrotaline (MCT) group compared to the control group at 4 weeks after MCT injection (Figure 4B), which was accompanied by an increase in the right ventricular hypertrophy index (RVHI) and the ratio of right ventricle (RV) weight to body weight in the MCT group (Figure 4C,D). Meanwhile, the H&E results demonstrated enhanced pulmonary arterial remodeling (PAR) in the MCT group (Appendix A), as indicated by the relative value of WT% and WA% (WT% = (the diameter of external vessel – the diameter of internal vessel)/the diameter of external vessel × 100%; WA% = (the total area of the vessel – the lumen area of the vessel)/the total area of vessel × 100%) (Appendix A), compared to the rats treated with saline. Consistently, similar results were observed in the PAH mouse model induced by chronic hypoxia treatment for 3 weeks (Figure 5 and Appendix A). PAR of rodents was also evaluated by immunohistochemistry and immunofluorescence (Appendix A), which was consistent with the results of the H&E. These data verified that the PAH models were successfully established, mainly characterized by means of RVSP and PAR.

### 2.5. Validation of Hub Genes

There were multiple transcripts of *TSHZ2*, and the sequence of each transcript was not verified according to the data obtained from National Center for Biotechnology Information [25]. Therefore, we could only focus on the mRNA expression levels of *PARM1* and *CCDC80*. RT-PCR revealed that the mRNA expression levels of *PARM1* and *CCDC80* were significantly different in MCT rat/hypoxia mouse lungs compared to the normal controls (Figure 4E,H; Figure 5E,F). Further RT-PCR of the extractions of the pulmonary arteries of rats in both groups showed the same result, indicating the vital roles of *PARM1* and *CCDC80* in the MCT model (Figure 4F,I). Furthermore, we detected a change in the mRNA levels of *PARM1* and *CCDC80* between hypoxia-exposed cells and normoxia-exposed cells. The expression levels of both *PARM1* and *CCDC80* mRNA were slightly elevated in the hypoxia group, in contrast to the normoxia group (Figure 4G,J). These results confirmed the importance of *PARM1* and *CCDC80* in PAH at the ex vivo and in vitro levels, respectively. Consistent with the above findings, Sasagawa et al. identified *CCDC80* as a novel gene related to PAH using a bioinformatic approach [20]; however, the role of *PARM1* in PAH has been hitherto unknown. Thus, we aimed to explore the potential contribution of *PARM1* to PAH in the present study.

### 2.6. PARM1-Mediated AKT/FOXO3A Pathway in PAH

PARM1 is a member of the mucin family with multiple domains, including signal peptide, extracellular domain, transmembrane domain, and cytoplasmic tail, and is associated with proliferation, differentiation, and apoptosis [26,27,28,29,30]. Charfi et al. identified PARM1 as a novel potential oncogene that could enhance NIH/3T3 cell proliferation via the AKT-dependent signaling pathways [26]. They found an upregulation of the phosphorylation state of AKT in cell lysates from NIH/3T3 cells overexpressing PARM1. Thus, we hypothesized that PARM1 could regulate PAH via the AKT/FOXO3A signaling pathway. As presented, the Western blot results of lung homogenates showed an increase in PARM1 in the PAH rodent group compared with the control group (Figure 6A,H and Figure 7A,H). Correlation analysis revealed that there was a positive linear correlation between the expression level of PARM1 protein and PCNA protein in the rat and mouse lung tissues (Figure 6A,B and Figure 7A,B). This suggested that PARM1 could be involved in cell proliferation. Further Western blot analysis revealed that the PAH rodents group, in contrast to the control group, had an elevated phosphorylated state of AKT (p-AKT/AKT), a crucial effector that responds to various growth factors, as well as upregulated phosphorylation of FOXO3A (p-FOXO3A/FOXO3A) (Figure 6C–G and Figure 7C–G), resulting in cell proliferation in the pulmonary artery, which is regarded as a prominent mechanism in the development of PAH. We previously reported that the level of serum 14-3-3β was associated with the severity of PAH [17]. In addition, several studies have demonstrated that phosphorylation of FOXO3a caused it to transfer from the nucleus to the cytoplasm, where it binds to 14-3-3 proteins, preventing it from re-entering the nucleus [31,32]. Thus, we detected the expression of 14-3-3β in the PAH rodent lung tissues; however, a significantly different expression level of 14-3-3β was not found between the control group and the PAH group (Appendix A). Taken together, these results indicated that PARM1 was activated in the PAH group and could promote PASMC proliferation via the AKT/FOXO3A signaling pathway in PAH. 

### 2.7. PARM1 Promotes Proliferation via the AKT/FOXO3A Pathway in PASMC under PDGF-Stimulated or Hypoxia Conditions In Vitro

To further confirmed our hypothesis, we implemented in vitro cell culture studies. Western blot analysis showed that PDGF/hypoxia could promote the expression of PARM1 in primary isolated rat PASMC (Figure 8A,B,H,I), which was consistent with our ex vivo findings. Moreover, increased phosphorylation of AKT and FOXO3A was detected during the PDGF-BB/hypoxia treatments (Figure 8A,C–F,H,J–M). Simultaneously, the level of PCNA was significantly upregulated in PASMCs treated with PDGF/hypoxia (Figure 8A,G,H,N). As anticipated, these effects induced by PDGF-BB/hypoxia were weakened by two independent siRNA for PARM1; namely, si-PARM1-01 and si-PARM1-02 (Figure 8A–N and Appendix A). To determine the exact biological role of PARM1, the effect of PARM1 silencing on the proliferation of PDGF/hypoxia-exposed PASMCs was explored. EdU assays indicated that the proliferation rates of PDGF/hypoxia-treated PASMCs were increased compared to that of the controls and were dramatically attenuated by PARM1 downregulation (Figure 8O–R).

In conclusion, we hypothesized that PARM1 could promote PASMC proliferation via an AKT/FOXO3-dependent signaling pathway in PAH, which is illustrated in Figure 9.

## 3. Discussion

Bioinformatics and omics tools are of benefit in investigating the causes, mechanisms, and outcomes of disease [33,34,35]. In this study, WGCNA was applied to construct co-expression networks and explore the gene expression in PAH, based on the GSE15197, GSE113439, GSE117261, and GSE53408 datasets; and the white module was found to be most positively correlated with PAH. The top hub genes, including *PARM1*, *CCDC80*, and *TSHZ2*, were identified in silico. We then performed RT-qPCR to verify the identified hub genes in the MCT-induced and hypoxia-induced PAH animal models, and PARM1 was selected for the subsequent analysis. The potential role of PARM1 in the process of PAH was further investigated in vitro and ex vivo. We demonstrated that PARM1 might promote the proliferation of PASMCs in the PAH lung via an AKT/FOXO3A/PCNA-dependent signaling pathway.

PARM1 was first discovered by Bruyninx et al., who verified that PARM1 exhibited a specific activity in the lower genitourinary tract and was expressed in an androgen-regulated manner [28]. Fladeby et al. further reported that rat PARM1 was overexpressed in the degenerative prostate gland after androgen withdrawal, while human PARM1 was downregulated in regressing human prostate cancer xenograft [29]. Intriguingly, they also found that PARM1-transfected cells displayed a significantly higher number of colonies than empty vector transfected cells, implicating a potential role of PARM1 in cell proliferation. The same function of PARM1 was discovered by Charfi et al. [26], which attracted our attention to the role of PARM1 in PAH. In addition, Charfi et al. hypothesized that PARM1-induced phosphorylation of AKT might be associated with cell proliferation. In the current study, we concentrated on the signaling pathways where PARM1 was involved, and we showed that PARM1 could induce PASMC proliferation via an AKT/FOXO3A/PCNA-dependent signaling pathway in PAH.

AKT is a serine/threonine protein kinase that can exert multiple functions, mainly through its key multifunctional downstream signaling nodes (GSK3, FOXO, mTORC1, etc.) [36]. Activation of phosphoinositide 3-kinase, a dominant upstream regulator of AKT, induces phosphoinositide-dependent protein kinase 1-mediated phosphorylation of Thr308 on AKT, which is required for AKT kinase activity [37,38]. Maximal activation of the kinase requires phosphorylation of Ser473 in the C-terminal hydrophobic motif [36]. Many studies have revealed that AKT is associated with numerous pathophysiological processes, including obesity, diabetes, cancers, and cardiovascular diseases [39,40,41,42,43]. Moreover, several researchers indicated that overactivated AKT was associated with vascular remodeling [44], which, as previously mentioned, is the basic mechanism underlying PAH. Indeed, Teng et al. demonstrated that inhibition of AKT^Thr308^ phosphorylation could reduce the proliferation of PASMCs and alleviate pulmonary arterial remodeling [45]. The results of the present study showed that the levels of phosphorylated AKT, both in the lung tissues of MCT/hypoxia-treated rodents and in PDGF/hypoxia -exposed PASMCs, were significantly increased compared to the corresponding controls. In addition, we also demonstrated that knockdown of PARM1 in PASMCs could inhibit the activation of AKT. 

Downstream of AKT, we found that FOXO3A could regulate cell proliferation in the rodent PAH models. FOXO3A, a member of the forkhead box class O (FOXO) family, plays an important role in controlling various cellular processes [46]. Zanella et al. illustrated that the function of FOXO3A depended on its subcellular localization [47]. AKT-mediated phosphorylation of FOXO3A^Ser253^ results in its translocation from the nucleus to the cytoplasm [46,48], where it associates with 14-3-3α/β, 14-3-3ζ, and 14-3-3σ, and this binding prevents FOXO3A from re-entering into the nucleus [49,50,51], resulting in deregulation of cell proliferation [52]. 

The 14-3-3 proteins are a collection of evolutionarily conserved proteins that regulate multiple biological processes, including cell cycle, cell proliferation, cell migration, apoptosis, autophagy, and gene transcription [53,54,55,56,57]. Seven 14-3-3 subtypes (α/β, γ, ε, η, σ, τ [also called θ] and ζ/δ) have been reported in mammals [58]. We have previously shown that there is a link between a high serum level of 14-3-3β and the severity of PAH [17]. Nevertheless, our present work indicated no significant difference in the expression of the 14-3-3β protein found in the lungs between the PAH group and the control group. These two results seem to contradict each other, but the level of 14-3-3β in serum did not reflect that in tissue. There are three possibilities that we could think of: 1. The elevated 14-3-3β proteins in the lung tissue are immediately secreted in the blood under PAH; 2. Species differences caused different results. The experiments in this study were based on rodent tissue and cells, while the previous paper was based on the validation of human blood samples; 3. The elevated levels of 14-3-3β in the serum might have been derived from the secretions of other tissues.

Several limitations should be acknowledged in this study. First, we only validated the identified hub genes in rodent models, not in human samples, limiting the clinical significance. Second, the study was based on the lung tissue specimen gene expression profile from GEO. As is well known, lung tissue is composed of various functional cells. A more accurate understanding of PAH pathophysiology could be provided using a single-cell transcriptome. Third, loss- or gain-of-function experiments for *PARM1* in vivo are required to support the role of PARM1 in PAH. 

## 4. Materials and Methods

### 4.1. Data Collection and Data Preprocessing

The workflow is illustrated in Figure 10. The gene Expression Omnibus (GEO) datasets were explored, and four datasets (Table 1) were selected to screen the potential markers of PAH. We combined the three microarray datasets of GSE15197, GSE113439, and GSE117261 with combat packages, and obtained 148 samples (99 PAH samples and 49 Controls), each containing 17,664 genes [59,60,61]. This task was completed by the “ComBat” tool from the R-package “sva”. GSE53408 was utilized as a validation set to verify the identified candidate genes [62]. All data were analyzed and visualized using R Software 4.1.0.

### 4.2. Differential Expression Gene (DEG) Selection

The R package “limma” was adopted to select the DEGs. Gene expression data were stabilized and normalized using the voom algorithm. Differential gene comparison was performed using the contrast function, and the empirical cut-off values for adjusting *p*-values < 0.05 and |log2 fold change| > 1 were screened for the DEGs of the fusion gene set with the “limma” packages.

### 4.3. Module Construction Using the R-Pack WGCNA

The R package WGCNA was applied to construct the module. In brief, WGCNA was performed for analysis of the co-expression network and to determine the hub genes. Gene network (module) construction was performed using a soft-threshold power of 6, and 42 modules were identified through hierarchical clustering. The biologically interesting modules were identified using the relation of the modules and the clinical traits. Gene significance (GS) is an important parameter to measure the relationship between modules and clinical traits and is defined by the minus log of a *p*-value.

After identifying the biologically interesting modules, we further explored the biological meanings of these modules. The biological meaning of a module was explored by applying the gene ontology (GO) enrichment analysis contained in the R-package ClusterProfiler. We used the R-package ggplot2 to visualize these results. The hub gene was defined as an abbreviation of “highly connected gene” selected through the construction of a co-expression network. We then verified hub genes in the validation dataset. The “ggraph” package was used to visualize the network.

### 4.4. Functional and Pathways Enrichment Analysis

The R package “ClusterProfiler” was used for GO enrichment analysis [63]. The reference database http://org.Hs.eg.db (accessed on 19 October 2021) provided the information for GO analysis with a *p*-value cutoff equal to 0.01.

### 4.5. Validation of Hub Genes

GSE53408, containing the lung tissue from 11 normal and 12 severe PAH patients, was extracted from the GEO database and utilized as a validation dataset. An unpaired Student’s *t*-test from R software (version 4.1.0) was used to contrast the intergroup gene expression variances. The diagnostic efficacy of the selected hub genes was assessed using the receiver operating characteristic (ROC) curve created using the “pROC” package in R. Decision curve analysis was performed by decision curve function in “rmda” package.

### 4.6. Animal Model of PAH

The animal study was carried out in accordance with procedures approved by the Institutional Animal Care and Use Committee of Tongji Medical College, Huazhong University of Science and Technology. Monocrotaline (MCT, Cat# C2401, Sigma-Aldrich, Saint Louis, MO, USA) was dissolved in 1 mol/L HCl, the pH was maintained to 7.4 with 10 mol/L NaOH, and the stock concentration was titrated with sterilized water to 10 mg/mL. Young male Sprague–Dawley (SD) rats (200–250 g, aged 6 weeks) were purchased from Hunan SJA Laboratory Animal Co., Ltd. (Hunan, China). The MCT-induced PAH rats received a single dose of 40 mg/kg (Intraperitoneal) MCT. The control group was injected with the same amount of saline instead of MCT [64]. For the hypoxia-induced PAH mouse model, we acquired male C57BL/6J mice (19–21 g, aged 8 weeks) from Vital River Laboratories (Beijing, China). Mice in the hypoxia group were reared in a closed chamber with 10% oxygen concentration for 21 days to induce PAH, while mice in the normoxia group were reared normally [65]. 

### 4.7. Hemodynamics Parameter Measurements and Assessment of Right Ventricular Hypertrophy

The experimental male rats or mice were anesthetized via intraperitoneal injection of pentobarbital sodium (30 mg/kg). Subsequently, we inserted a Millar Mikro-Tip catheter (SRP-671, AD Instruments, Dunedin, New Zealand) from the right jugular vein into the right ventricle, to obtain the right ventricle systolic pressure (RVSP) of all the rats. To acquire the RVSP of mice, we opened the thoracic cavity and directly inserted the catheter into the right ventricle. The right ventricular hypertrophy index (RVHI), which is the ratio of the weight of the right ventricular and that of the left ventricular plus ventricular septum [66,67], and the ratio of the right ventricle weight to body weight were used to evaluate the right ventricle hypertrophy.

### 4.8. Cell Culture and Treatment

Male SD rats aged 4 weeks were sacrificed to harvest the lungs, from which the second or third pulmonary arteries were separated. Next, we tore out the outer membrane of the vessel, slit the vessel longitudinally, and wiped the inner membrane with a sterile cotton swab. Primary PASMCs were isolated from the generated media and then cultured in DMEM supplemented with 20% fetal bovine serum, 100 U/mL penicillin, and 0.1 mg/mL streptomycin at 37 °C aerated with 5% CO_2_ in a humidified incubator [68]. There was no additional growth supplement added to the medium. Only cells between passage three to eight were used in our study.

Specific small interfering RNA (siRNA) was purchased from Guangzhou RiboBio Co., Ltd. (Guangzhou, China). The si-PARM1-01 sequence was GCTCTACTGTGAACAACAT, and the si-PARM1-02 sequence was ACAACAACCCTCTCTATGA. The PASMCs were seeded in six-well plates and then transfected with 60 nM siRNA targeting PARM1 (si-PARM1) or negative control siRNA using Lipofectamine^TM^ 2000 reagent (Cat# 11668019, Invitrogen, Carlsbad, CA, USA), in accordance with the manufacturer’s instructions. Another batch of PASMCs was treated with recombinant human PDGF-BB protein (30 ng/mL) (PeproTech, Cat# 100-14B, Suzhou, China) or using a hypoxia incubator (1% O_2_, 5% CO_2_, balance N_2_) (Billups-rothenberg, San Diego, CA, USA).

### 4.9. Reverse-Transcriptional Quantitative Polymerase Chain Reaction (RT-qPCR)

RT-qPCR was performed as previously reported [16,69]. Briefly, total RNA was extracted from the frozen lung tissues, separated pulmonary arteries, and cultured PASMC using TRIzol reagent (Takara, Kyoto, Japan). Then, 1000 ng RNA was reversely transcribed into complementary DNA using ABScript III RT Master Mix for qPCR with gDNA remover (Abclonal, Wuhan, China). The quantitative real-time polymerase chain reaction was formulated using 2× Universal SYBR Green Fast qPCR Mix (Abclonal, Wuhan, China). The 2^−ΔΔCt^ method was adopted to calculate the relative gene expression, and β-actin served as an internal control for normalization. The primer sequences of these genes were as follows:

Rat:

*PARM1*: F: 5′-TGGCACGGTTGTGTCTTTCT-3′, R: 5′-TCCAGAGATGACTGTGAGTTTACC-3′; *CCDC80*: F: 5′-ACCAGGAGGCCTAACAAAGC-3′, R: 5′-TCGTTTGTCTGTCCGGTTGT-3′.

Mouse:

*PARM1*: F: 5′-TCACAACGCCTCAGTTCTCC-3′, R: 5′-GGCTGGTGACAGCTTCTTCT-3′; *CCDC80*: F: 5′-GTCAGCTCGGTATCTGAGGC-3′, R: 5′-TCTCATGGTCGGGTGAGCTA-3′.

### 4.10. Western Blotting

Lung tissues and cultured PASMCs were lysed with ice-cold RIPA lysis buffer containing phosphatase and protease inhibitor cocktail (MedChemExpress, Monmouth Junction, NJ, USA). A BCA assay (Boster, Pleasanton, CA, USA) was employed for determining the extracted protein concentration. Equal tissue or cell lysates were resolved using 10% SDS-polyacrylamide gel electrophoresis (PAGE) and were subsequently transferred into PVDF membranes (Merck Millipore, Tullagreen, Carrigtwohill, County Cork, Ireland). The blots were blocked for 1 h in 5% bovine albumin (BSA) in Tris-Buffered Saline with Tween 20 (TBST) and then incubated at 4 °C overnight with appropriate primary antibodies. The antibodies used were as follows: PARM1 (1:2000, Cat# ab168369, Abcam, Cambridge, UK), AKT (1:1000, Cat# A18675, Abclonal), P-AKT T308 (1:500, Cat# sc-271964, Santa Cruz), P-AKT S473 (1:1000, Cat# 4060 s, Cell Signaling, 3 Trask Lane Danvers, MA, USA), FOXO3A (1:1000, Cat# 66428-1-Ig, Proteintech, Wuhan, China), P-FOXO3A S253 (1:1000, Cat# AP1131, Abclonal, Wuhan, China), P-FOXO3A S294 (1:1000, Cat# AP0856, Abclonal, Wuhan, China), PCNA (1:1000, Cat# A0264, Abclonal, Wuhan, China), and β-actin (1:5000, Cat# 66009-1-Ig Proteintech, Wuhan, China). Afterwards, the membranes were washed and incubated for 1 h with HRP-conjugated anti-mouse IgG (Cat# 111-035-003, Jackson ImmunoResearch, West Grove, PA, USA) or anti-rabbit IgG (Cat# 115-035-003, Jackson ImmunoResearch, West Grove, PA, USA) secondary antibodies at room temperature (RT). The Western blot images were captured using a Tanon-5200 Chemiluminescent Imaging System (Tanon Science & Technology Co., Ltd., Shanghai, China).

### 4.11. Histology and Immunohistochemistry

The lung tissues were cautiously harvested, fixed with 4% formaldehyde overnight, paraffin-embedded, and then cut into 5 μm thick slices. Then the lung slides were deparaffinized, rehydrated, and boiled in 5% citric acid buffer (pH = 6.0) for 20 min for antigen retrieval. Subsequently, we blocked slides with 5% BSA in TBST for 1 h and incubated them overnight with primary antibodies against α-SMA (1:100, Cat# GB13044, Servicebio, Wuhan, China) at 4 °C, followed by the corresponding secondary antibodies (1:200) for 30 min at RT. The staining result was visualized with 3,3′-diaminobenzidine (Vector Laboratories, Burlington, CA, USA) and counterstained with hematoxylin under a light microscope (Olympus BX61, Tokyo, Japan) [70]. Hematoxylin and eosin (H&E) staining of the lung slides was performed for morphological analysis. Pulmonary vascular remodeling indices (WT% and WA%) were analyzed using Image J, to determine the medial wall thickness. WA% and WT% were calculated using the ratio of the vessel wall area to the total vessel area ((the total area of the vessel−the lumen area of the vessel)/the total area of the vessel × 100%) and the ratio of the vessel wall thickness to the vessel diameter ((the diameter of the external vessel−the diameter of the internal vessel)/the diameter of the external vessel × 100%), respectively [71,72]. 

### 4.12. Immunofluorescence

Deparaffinized lung sections were boiled in citric acid buffer (pH = 6.0) for 20 min to recover antigens. After cooling, the sections were blocked with blocking solution (5% BSA in TBST) for 1 h and incubated at 4 °C overnight with primary antibodies against α-SMA (1:100, Cat# GB13044, Servicebio, Wuhan, China), followed by incubation with a goat anti-mouse-Cy3 (1:200, Cat# GB21301, Servicebio, Wuhan, China) for 1 h at RT. Slides were washed with PBS three times, followed by counterstaining with 4′,6-diamidino-2-phenylindole fluorescent dye (DAPI). Digital images were examined under a microscope using a 20× objective lens with SOPTOP ICX41(Yuyao, China).

### 4.13. Cell Proliferation Assay

Cell proliferation assays were carried out using a CellorLab™ EdU Cell Proliferation Kit with Alexa Fluor 488 (Epizyme, Shanghai, China) according to the manufacturer’s protocol. Following treatment, PASMCs were treated with 10 μmol/L EdU for 2 h at 37 °C. Afterwards, cells were fixed and permeabilized at RT. After being washed three times, the cells were incubated with Click additive solution at RT away from light for 30 min. Finally, the nuclei of all cells were stained with 1× Hoechst 33342. Three random microscopic images were selected for further analysis. The extent of PASMCs proliferation was determined using the cell proliferation rate, which was calculated as the ratio of the number of EdU-positive cells (green fluorescence) to that of all cells (blue fluorescence).

### 4.14. Statistical Analysis

All data were presented as mean ± standard error of the mean (SEM). Data analyses were executed with the statistical program GraphPad Prism (version 9.0, San Diego, CA, USA). Correlations between the expression levels of two proteins were analyzed using linear regression. The predictive values of the hub genes were evaluated with ROC analysis. The detailed statistical analysis of bioinformatics part was described above (Section 4.2, Section 4.3 and Section 4.4). Student’s *t*-test was applied for statistical analyses, and differences at the 95% confidence level (*p* < 0.05) were considered statistically significant.

## 5. Conclusions

In summary, we identified PARM1 as a vital hub gene in PAH through WGCNA. The results of our study suggest that PARM1 might be a promising biomarker and feasible therapeutic target for PAH.

## Figures and Tables

**Figure 1 ijms-24-06385-f001:**
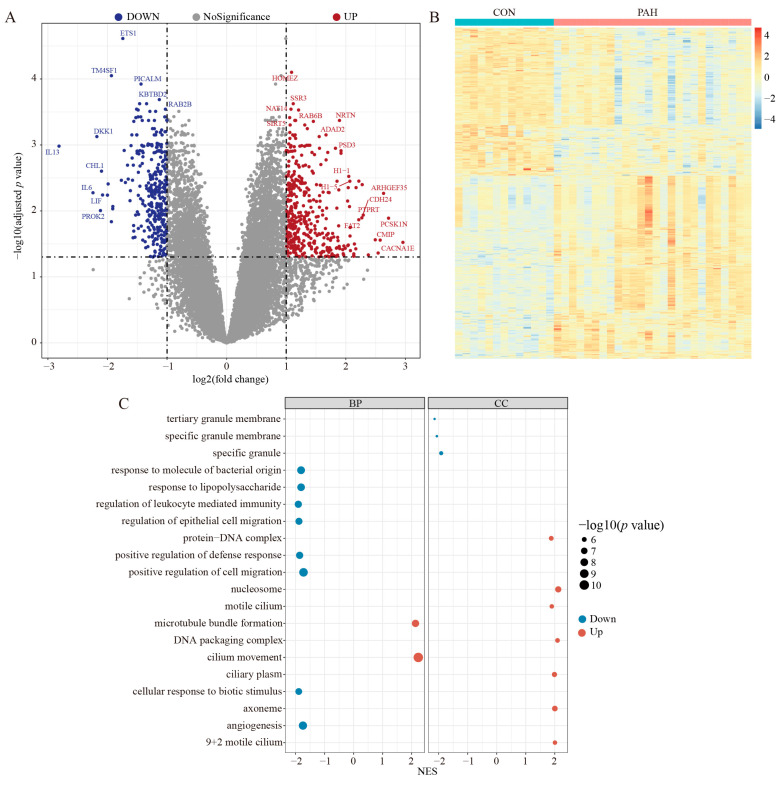
Differential expression gene analysis of PAH and control patients. (**A**) Volcano plots of DEGs, up means the upregulated genes in PAH patients and down means downregulated in PAH patients when compared with normal lung tissue. (**B**) Heat map of the DEGs expressions of normal and PAH lungs. (**C**) GO gene set enrichment analysis of the DEGs with *p* values < 0.01. NES: normalized enrichment score; DEGs: differential expression genes.

**Figure 2 ijms-24-06385-f002:**
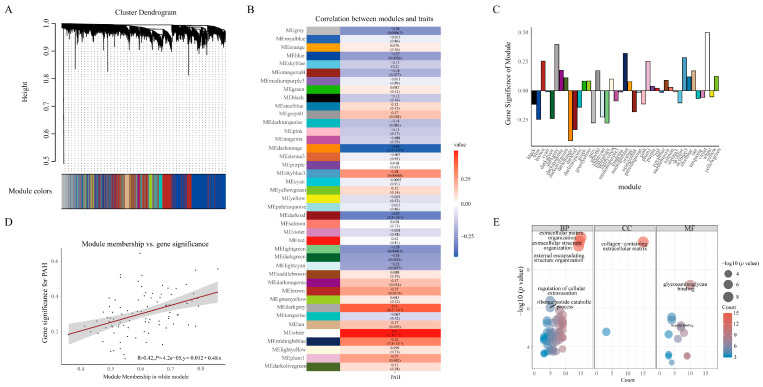
The module detection with WGCNA. (**A**) Clustering dendrogram plot of PAH and normal lungs. (**B**) Heat map of the correlation between modules and traits. (**C**) Gene significance (GS) bar plot of modules. (**D**) Module membership (MM) and the gene significance (GS) correlation plot. (**E**) GO enrichment analysis of white module genes. BP: biology pathways; CC: cell component; MF: molecular function.

**Figure 3 ijms-24-06385-f003:**
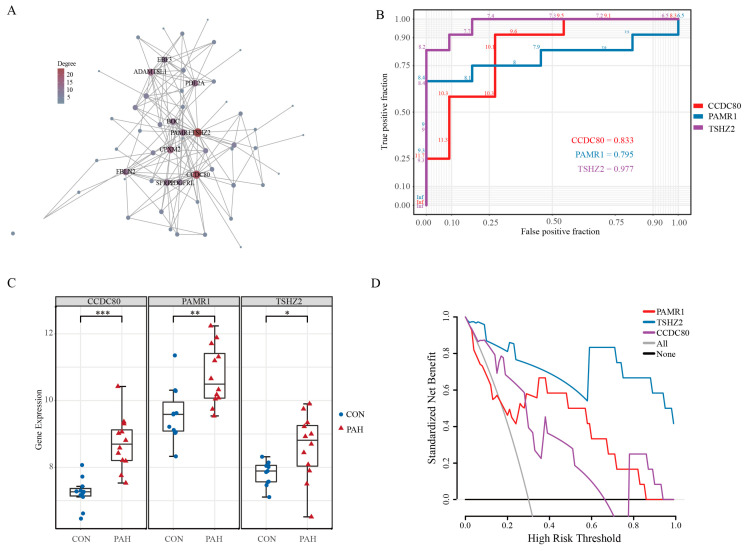
Hub gene selection of white module. (**A**) Network of the white module. (**B**) ROC curve of *CCDC80*, *PAMR1*, and *TSHZ2*. (**C**) The gene expression of *CCDC80*, *PAMR1*, and *TSHZ2* in the validation dataset (GSE53408). (**D**) Decision curve analysis for the *PAMR1*, *TSHZ2*, and *CCDC80* risk prediction models. *p*-value was calculated using Student’s *t*-test analysis. * *p* < 0.05, ** *p* < 0.01, *** *p* < 0.001 vs. CON group (student’s *t*-test). *CCDC80*: coiled-coil domain containing 80, *PAMR1*: peptidase domain containing associated with muscle regeneration 1, *TSHZ2*: teashirt zinc finger homeobox 2.

**Figure 4 ijms-24-06385-f004:**
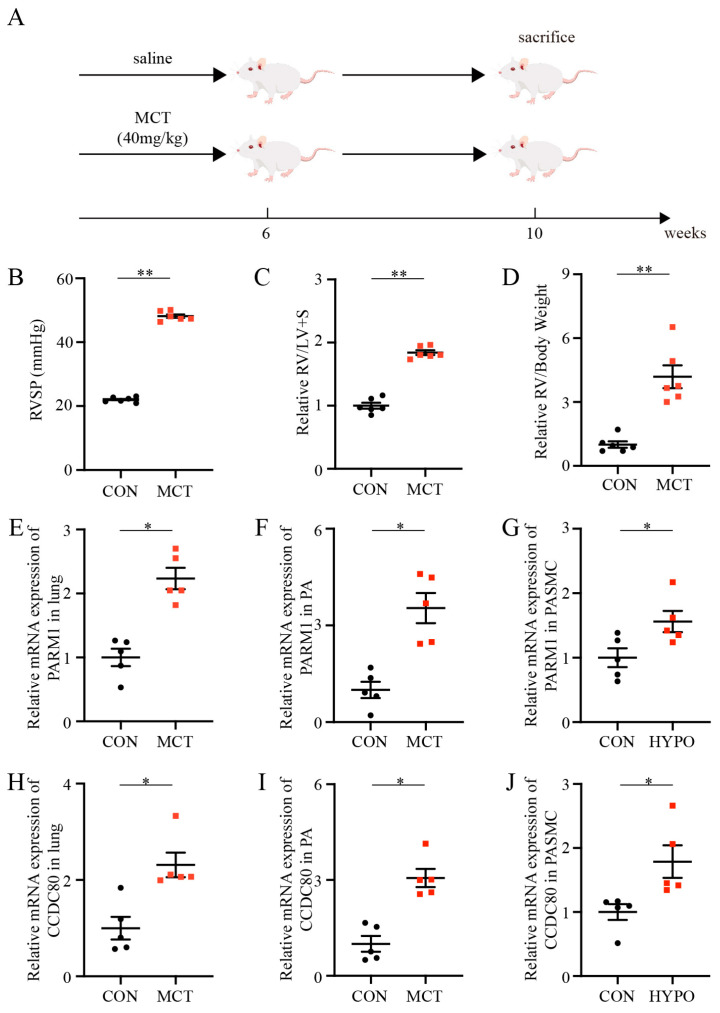
Establishment of pulmonary arterial hypertension rat model and hub gene validation. (**A**) Schematic illustration of pulmonary arterial hypertension rat model. (**B**–**D**) RVSP, RV/(LV + S), and RV/body weight at 4 weeks after MCT injection (n = 6 each). (**E**–**J**) Relative mRNA expression of *PARM1* and *CCDC80* normalized to a β-actin internal control (n = 5 each). Data are shown as mean ± SEM. * *p* < 0.05, ** *p* < 0.01 vs. CON group (student’s *t*-test). CON, control rats without pulmonary arterial hypertension; PAH, pulmonary arterial hypertension; MCT, monocrotaline; RVSP, right ventricle systolic pressure; RV, right ventricle; LV, left ventricle. S, septum.

**Figure 5 ijms-24-06385-f005:**
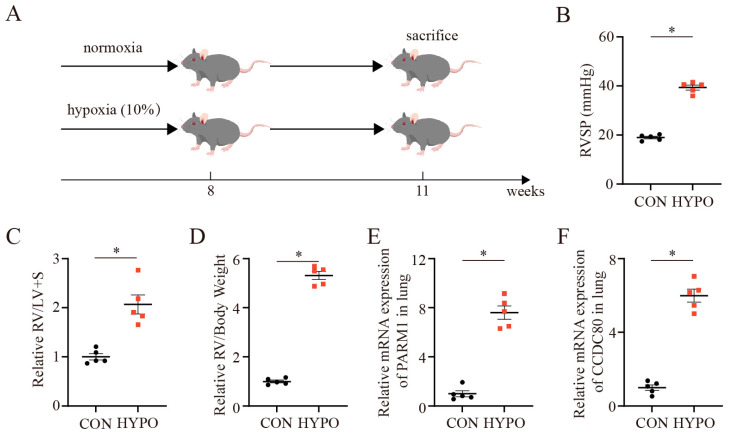
Establishment of pulmonary arterial hypertension mouse model. (**A**) Schematic illustration of pulmonary arterial hypertension mouse model. (**B**–**D**) RVSP, RV/(LV + S), and RV/body weight at 3 weeks after chronic hypoxia exposure (n = 5 each). (**E**,**F**) Relative mRNA expression of *PARM1* and *CCDC80* normalized to a β-actin internal control (n = 5 each). Data are shown as mean ± SEM. * *p* < 0.05 vs. CON group (student’s *t*-test). CON, control mice without pulmonary arterial hypertension; HYPO, hypoxia; RVSP, right ventricle systolic pressure; RV, right ventricle; LV, left ventricle. S, septum.

**Figure 6 ijms-24-06385-f006:**
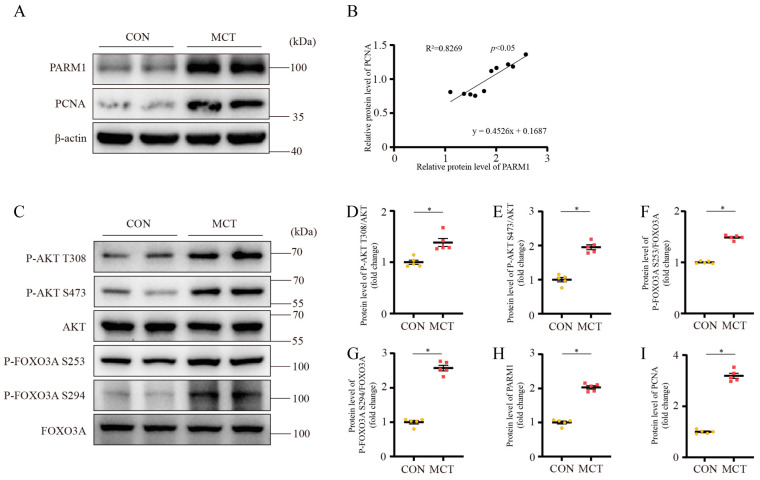
Screening of a novel therapeutic target for the PAH rat model. (**A**,**H**,**I**) Representative Western blots and analysis of PARM1 and PCNA expression in rat lung. Equal protein loading was confirmed using an anti-β-actin antibody (n = 5 each). (**B**) Linear regression analysis of the association between PARM1 and PCNA expression in rat lung. (**C**–**G**) Representative Western blots and analysis of p-AKT(T308)/AKT ratio; p-AKT(S473)/AKT ratio; p-FOXO3A(S253)/FOXO3A ratio; p-FOXO3A(S294)/FOXO3A ratio in rat lung. Data are shown as mean ± SEM (n = 5 each); * *p* < 0.05 vs. CON group (student’s *t*-test).

**Figure 7 ijms-24-06385-f007:**
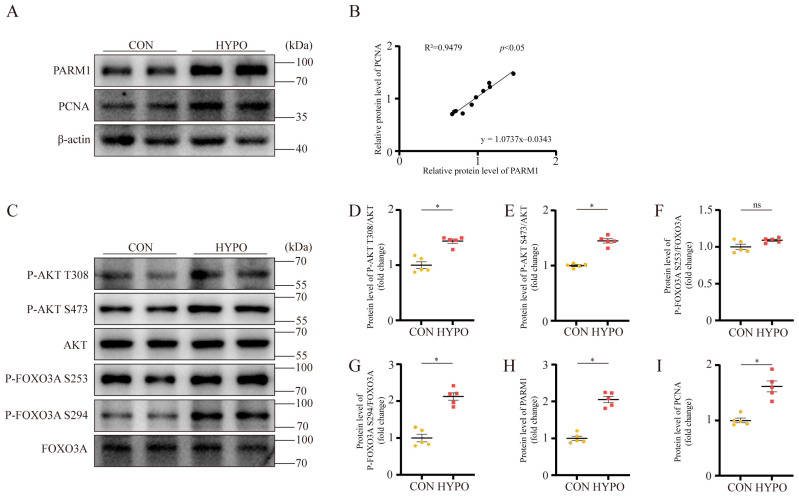
Validation of the screened hub gene in hypoxia-induced PAH mouse model. (**A**,**H**,**I**) Representative Western blots and analysis of PARM1 and PCNA expression in mouse lung. Equal protein loading was confirmed using an anti-β-actin antibody (n = 5 each). (**B**) Linear regression analysis of the association between PARM1 and PCNA expression in mouse lung. (**C**–**G**) Representative Western blots and analysis of p-AKT(T308)/AKT ratio; p-AKT(S473)/AKT ratio; p-FOXO3A(S253)/FOXO3A ratio; p-FOXO3A(S294)/FOXO3A ratio in mouse lung. Data are shown as mean ± SEM (n = 5 each); * *p* < 0.05 vs. CON group; ns represents “not significant” (student’s *t*-test).

**Figure 8 ijms-24-06385-f008:**
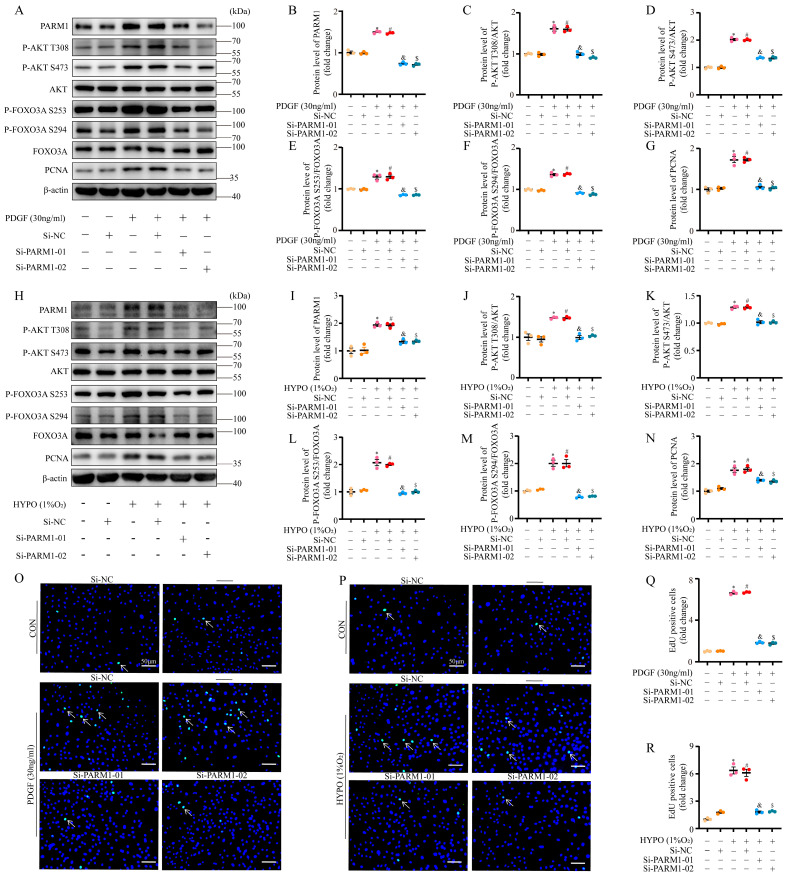
PARM1 siRNA exhibited the anti-proliferation effects in PDGF/hypoxia-treated PASMCs. (**A**–**G**) Representative Western blots and quantification of PARM1, p-AKT(T308), p-AKT(S473), AKT, p-FOXO3A(S253), p-FOXO3A(S294), FOXO3A, and PCNA in cultured PASMCs under control or PDGF (30 ng/mL) conditions, plus treatment with si-RNA targeting PARM1 (si-PARM1) or negative control siRNA (si-NC). Equal protein loading was confirmed using an anti-β-actin antibody. (**H**–**N**) Representative Western blots and quantification of PARM1, p-AKT(T308), p-AKT(S473), AKT, p-FOXO3A(S253), p-FOXO3A(S294), FOXO3A, and PCNA in cultured PASMCs under control or hypoxia (1% O_2_) conditions, plus treatment with si-PARM1 or si-NC. Equal protein loading was confirmed using an anti-β-actin antibody. (**O**,**Q**) Representative EdU incorporation assay and quantification of the ratio of PASMCs incorporated with EdU under the control or PDGF (30 ng/mL) condition plus treatment with si-PARM1 or si-NC, in which nuclei stained with 1 X Hoechst 33342 (blue) and incorporated EdU merged with 1× Hoechst 33342 are shown in green (n = 3 random microscopic visions; Scale bar, 50 μm). (**P**,**R**) Representative EdU incorporation assay and quantification of the ratio of PASMCs incorporated with EdU under the control or hypoxia (1% O_2_) conditions, plus treatment with si-PARM1 or si-NC, in which nuclei stained with 1× Hoechst 33342 (blue) and incorporated EdU merged with 1× Hoechst 33342 are shown in green (n = 3 random microscopic visions; Scale bar, 50 μm). Data are shown as mean ± SEM; * *p* < 0.05 vs. CON group; ^#^ *p* < 0.05 vs. the CON + SiNC group; ^&^ *p* < 0.05 vs. the HYPO/PDGF + SiNC group; ^$^ *p* < 0.05 vs. the HYPO/PDGF + SiNC group (student’s *t*-test). HYPO, hypoxia; PDGF, PDGF-BB.

**Figure 9 ijms-24-06385-f009:**
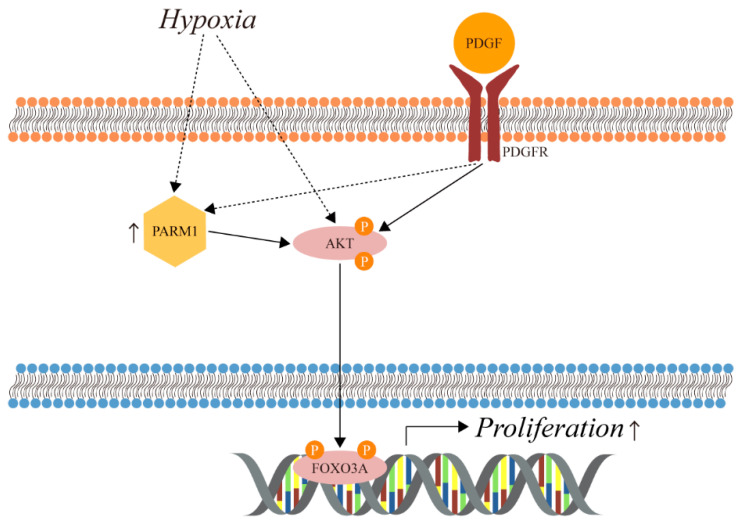
Schematic illustration shows PARM1 promoting proliferation in the pulmonary arterial hypertension via the AKT/FOXO3A pathway in PASMC. P signifies phosphorylation.

**Figure 10 ijms-24-06385-f010:**
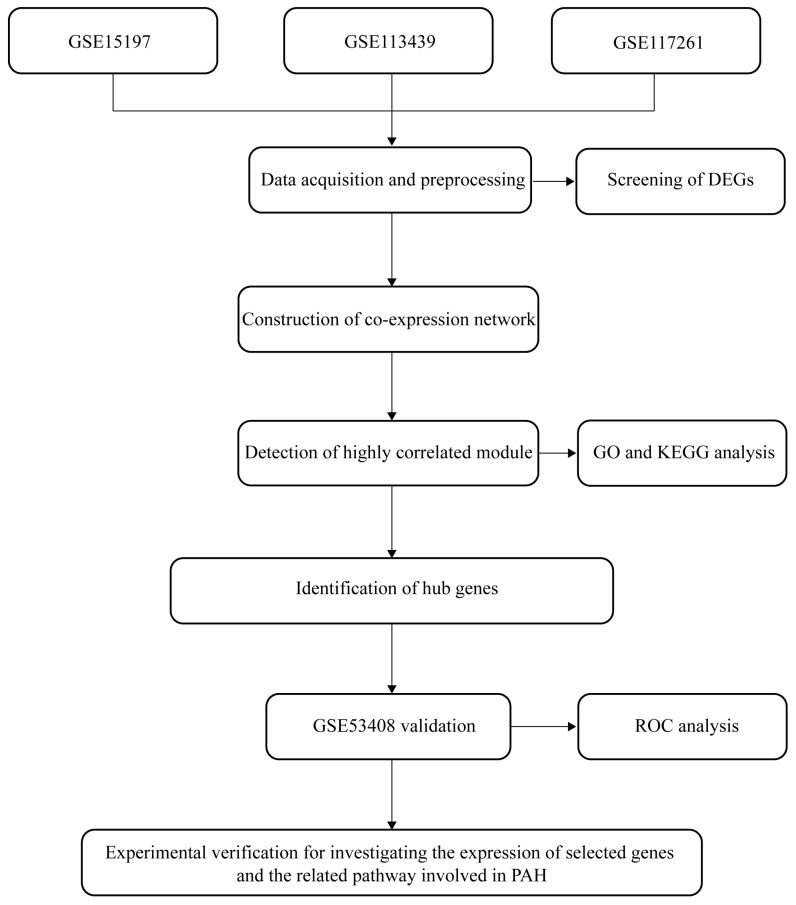
Workflow of the analysis procedure. Data collection, preprocessing, analysis, and validation. DEG, differentially expressed gene; GO, gene ontology; KEGG, Kyoto Encyclopedia of Genes and Genomes; ROC, receiver operating characteristic curve; PAH, pulmonary arterial hypertension.

**Table 1 ijms-24-06385-t001:** Gene expression dataset information in the GEO database.

Dataset	Spices	Organization	Datatype	PAH	Control
GSE15197	human	Lung	Expression data	26	13
GSE113439	human	Lung	Expression data	15	11
GSE117261	human	Lung	Expression data	58	25
GSE53408	human	Lung	Expression data	12	11

## Data Availability

All data that support the findings of this study are available from the corresponding author upon reasonable request.

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
