# Peer review of "PARM1 Drives Smooth Muscle Cell Proliferation in Pulmonary Arterial Hypertension via AKT/FOXO3A Axis"

_ijms, 2023, doi:10.3390/ijms24076385_

Round 1

Reviewer 1 Report

The study is well designed and executed. A few minor questions from this reviewer:

1) PAH animal models, such as the MCT-induced rat model or hypoxia-induced mouse model, are both well established. It is good that the authors showed a nice characterization of the two models in terms of PAH, but some of the characterizations can be included in the supplement files.

2) in figure 8, some of the annotations of statistical significance are not described in the figure legend. 

3) the si-PARM1 did not seem to knock down PARM1 expression significantly compared to the baseline. The reviewer is concerned about the actual efficiency and function of si-PARM1. Could authors provide more information or explanation on the si-PARM1 and their effects? 

Author Response

Comment 1:

PAH animal models, such as the MCT-induced rat model or hypoxia-induced mouse model, are both well established. It is good that the authors showed a nice characterization of the two models in terms of PAH, but some of the characterizations can be included in the supplement files

Answer 1:

We fully appreciate this reviewer’s suggestion. We have now re-organized the figures. Part of the general characterization of the two PAH models, such as H&E, αSMA immunohistochemistry and immunofluorescence results in Figure 4/Figure 5 were moved to the supplementary files Figure S2/Figure S3.

Comment 2:

In Figure 8, some of the annotations of statistical significance are not described in the figure legend.

Answer 2:

We apologize for the oversight. We have check it carefully and described all the annotations of statistical significance in the figure legend of the revised manuscript. As you can see, we described the meaning of all the annotations of statistical significance in the figure legends corresponding to Figure 8.

Comment 3:

The si-PARM1 did not seem to knock down PARM1 expression significantly compared to the baseline. The reviewer is concerned about the actual efficiency and function of si-PARM1. Could authors provide more information or explanation on the si-PARM1 and their effects?

Answer 3:

We appreciate the thoughtful comment of the reviewer. Actually, we had checked the knockdown efficiency of si-PARM1 before we performed formal experiments. We demonstrated that both PDGF-BB and hypoxia could upregulate the expression of PARM1, so it makes sense that the level of PARM1 of PASMCs treated with PDGF-BB/hypoxia and si-PARM1 is not significantly lower than that of the control. To show the actual knockdown efficiency of si-PARM1, we include the WB and qPCR results in the supplementary files Figure S5.

For detailed point to point response, please see the response to the reviewers document in the file system.

Reviewer 2 Report

1. The original paper of used GEO datasets needs to be cited.

2. In the lines 294-295, the DEGs were screened with the cut-off values of adjusting p-values < 0.05 and |log2 change 295 fold |> 0.05, and it seems too lax.

3. In then line 314, t-test can only be used if normal distribution and homogeneity of variance were satisfied.

4. In the present study, MCT-induced rat model and hypoxia induce mouse model were used. And the RVSP values of the two model were similar, but the degree of pulmonary vascular remodeling of hypoxia mouse was significantly less than that in MCT rats, as shown in HE staining and α-SMA immunofluorescence.

5. The cell culture conditions for PASMCs need to be shown, and whether smooth muscle cell growth supplement was added. 

6. The parameters of the hypoxia incubator for cell need to be described.

7. In the lines 191-193, the present results cannot supported this conclusion without loss or gain of function experiments, and the rescue experiment is of significance.

8. Most studied were performed using cell culture, which is not really reflecting PAH. In vivo evidence of PARM1-mediated PAH may make more sense.

9. As mentioned in Disscusion section, the expression of PARM1 can be influenced by the level of sex hormone. Wthere were sex differences in the role of PARM1 in PAH.

Author Response

Dear Editor,

Please find enclosed a revised version of the manuscript ijms-2072878(International Journal of Molecular Sciences)with the title “PARM1 Drives Smooth Muscle Cell Proliferation in Pulmonary Arterial Hypertension via AKT/FOXO3A axis”. 

We were pleased by the editors’ and reviewers’ overall positive comments and their constructive suggestions for the improvement of our manuscript. We included changes and new data based on the reviewers’ comments within the revised version. We feel that the new data generated following the suggestions supports our conclusions and strengthens the manuscript. The changes addressing the points raised by the reviewers are marked yellow in the current version.

Moreover, according to the suggestion, we rephrased some of the paragraphs and added proper references.

If accepted for publication, we would provide a finalized version of the manuscript with all these changes incorporated and marked text-passages cleared.

Appended is our detailed point-by-point response to the reviewers’ comments.

We are looking forward to hearing from you.

Sincerely,

Hesong Zeng

and

Hongjie Wang

Reviewer 3 Report

The manuscript entitled "PARM1 Drives Smooth Muscle Cell Proliferation in Pulmonary Arterial Hypertension via AKT/FOXO3A axis" by He et. al. demonstrated the logical reasoning behind their hypothesis that PARM1 could promote PASMCs proliferation via an AKT/FOXO3-dependent signaling pathway in PAH through in-silica analysis combined with in-vitro and in-vivo validations. As a reviewer and a potentially targeted reader of the journal, I have a few words to say.

I have been working as a reviewer for a while, but the present research done by He et. al. is just my new best study ever seen in the MDPI system. In a nutshell, I firmly believe that this solid study has the potential to be one of the best papers in this journal, if not solely the best one.

Essentially I do not have major comments for the manuscript, but the following articles are recommended to be taken as references to enhance the discussion part as they offer significant insights into the field of omics study:

1. https://www.frontiersin.org/articles/10.3389/fmolb.2022.834593/full 

2. https://www.frontiersin.org/articles/10.3389/fimmu.2022.1022147/full 

3. https://www.frontiersin.org/articles/10.3389/fimmu.2022.978865/full 

Furthermore, I noticed that a ROC analysis was employed. I might have missed something, but in my opinion, it would add some more strength if the authors do a Decision Curve Analysis as well.

Author Response

Comment 1:

Essentially, I do not have major comments for the manuscript, but the following articles are recommended to be taken as references to enhance the discussion part as they offer significant insights into the field of omics study:

  1. https://www.frontiersin.org/articles/10.3389/fmolb.2022.834593/full 
  2. https://www.frontiersin.org/articles/10.3389/fimmu.2022.1022147/full 
  3. https://www.frontiersin.org/articles/10.3389/fimmu.2022.978865/full 

Answer 1:

We thank this reviewer for recommending these articles as references. After we carefully read them, we believe it is just right to cite them to strengthen the discussion part.

As you can see at the begin of the discussion part (Page 14, line 210 to line 211), we added “Bioinformatics and omics tools benefit us in investigating causes, mechanisms and outcomes of diseases” and the above references.

Comment 2:

Furthermore, I noticed that a ROC analysis was employed. I might have missed something, but in my opinion, it would add some more strength if the authors do a Decision Curve Analysis as well.

Answer 2:

We thank this reviewer for this important comment. We totally agree with it. We have performed a Decision Curve Analysis and incorporated it in Figure 3D in the revised manuscript.

For detailed point to point response, please see the response to the reviewers document in the file system.

Reviewer 4 Report

The study focused on identifying novel therapeutic target in PAH using bioinformatics approach. Author described methodology well and discussed the results. The statistical methods were well justified. I have following concern-

  1. The authors used 3 microarray datasets GSE15197, GSE113439, and GSE117261. All datasets are from lung tissues, but 1st dataset is for Idiopathic PAH others are simply from PAH. Do authors think, is this rational to combine them?
  2. Did the authors perform the batch analysis manually or used some tool? Please briefly explain the tool for batch normalization.
  3. Lines 294-295 mentioned log2 change fold 0.05. Is this a typo error?
  4. Line 77-78: All three datasets were integrated. Is there any batch effect correction performed? If so, how?
  5. Line 152: Be specific which dataset is referred here.

Round 2

Reviewer 2 Report

Thanks to the author for thier replies, and my doubts have been answered.

Reviewer 4 Report

Thanks for considering the suggestions and revising the manuscript. It looks improved now.